# Studying the Water Supply System of the Roman Villa of Pisões (Beja, Portugal) Using Ground-Penetrating Radar and Geospatial Methods

Rui Jorge Oliveira [1,2,3,*] , Pedro Trapero Fernández [1,3,4,5,6] , Bento Caldeira [1,2,3] , José Fernando Borges [1,2,3] and André Carneiro [5,6]

1   Instituto de Ciências da Terra (ICT), Universidade de Évora, 7004-516 Évora, Portugal
2   Departamento de Física, Escola de Ciências e Tecnologia, Universidade de Évora, 7002-554 Évora, Portugal
3   Earth Remote Sensing Laboratory (EaRSLab), Universidade de Évora, 7002-554 Évora, Portugal
4   Área de Historia Antigua, Departamento de Historia, Geografía y Filosofía, Universidad de Cádiz, 11003 Cádiz, Spain
5   Centro de História de Arte e Investigação Artística (CHAIA), Universidade de Évora, 7002-554 Évora, Portugal
6   Departamento de História e Arqueologia, Escola de Ciências Sociais, Universidade de Évora, 7002-554 Évora, Portugal
*   Correspondence: ruio@uevora.pt

**Abstract:** The Roman villa of Pisões (Beja, Portugal) was part of the Lusitanian colony of Pax Iulia. This place stands out for the predominance of the water element in several structures of the villa, highlighting the *balneum* and the large *natatio*, one of the largest known in Roman Hispania. The records of the initial excavations that took place beginning in 1967 do not allow the establishment of clear functionalities of the villa. The University of Évora, the owner of the site, conceived an action plan for the requalification and enhancement of the archaeological site. One of the tasks aims to investigate the site using applied geophysics. This work analyses the landscape directly related to the villa, given that it is in the flooded area of a river with a Roman containment dam. It is uncertain whether the water supply comes from this structure or other nearby springs. The use of ground-penetrating radar, combined with unmanned aerial vehicles, all integrated in a geographic information system, allows us to determine the location of underground water connections and create a topographic model with high resolution. Considering all the information, we propose a model for water transport inside the villa and estimate the location of the water supply.

**Keywords:** GPR survey; Roman villa of Pisões; water supply location; combined archaeological data; Roman Lusitania

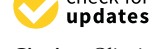



## 1. Introduction

The management and supply of water in a productive and recreational center, such as a Roman villa, was one of the needs that every owner had to fulfill. Generally, a water supply was required for human consumption and cooking, which came from wells or salubrious springs. A larger quantity was used for irrigated agriculture or for animal consumption, which may have come directly from streams and rivers, but did not require the same quality as drinking water. Therefore, there was high use of water for the recreation of the owners, generally in fountains, pools, and thermal baths. Studying how this water circuit worked, where the water collection points were, how it was distributed, and in what way requires a broad knowledge of Roman architecture and hydraulics, which is difficult to achieve in archaeological cases due to the lack of studies and precise contexts. In this regard, there are few cases of similar studies, especially concerning the involvement of non-invasive methodology or Geographic Information System (GIS) analysis of archaeological

remains [1,2]. There are cases of hydraulic exploitation, but they are focused on large architectural complexes and urban supply, such as Roman cities [3].

This work describes a case study of the Roman villa of Pisões, classified as a property of public interest in 1970, located in Santiago Maior (Figure 1), 10 km southwest of Beja.

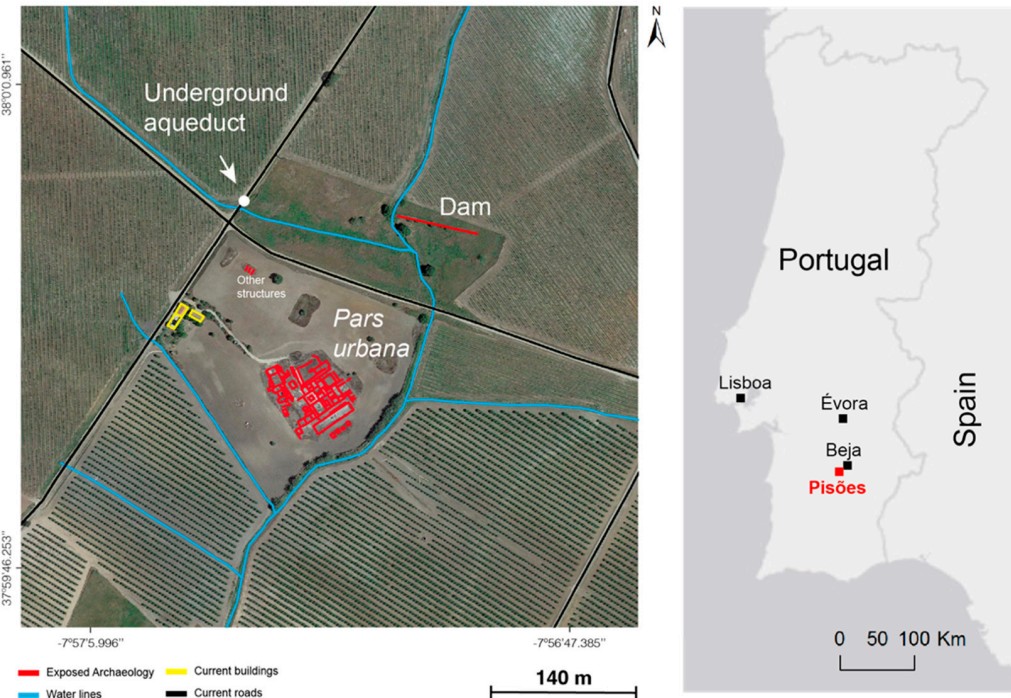

**Figure 1.** Location of the Roman villa of Pisões and general overview of the site.

The archaeological remains of the villa complex extend well beyond the protection area of approximately 6 ha divided into two adjacent areas. In the eastern area, there is a Roman dam [4], while in the western part, the center of the villa is located. Although the remains areas have been extensively excavated and the main results have been published, the reality is that Pisões was found by chance in 1967 when a tractor pulled a mosaic out of the ground. The excavation began immediately, coordinated by Fernando Nunes Ribeiro, one of the benefactors of the regional museum of Beja and, at the time, mayor of the municipality [5]. Although he had a great archaeological experience, he was not an archaeologist, but a veterinarian, which meant that the excavation was carried out without the appropriate rigor and methodology. For example, we do not know for sure when the foundation phase of the villa was built, although from the planimetry we can assume that it was sometime in the 1st century AD. Likewise, we do not know the duration of the occupation or when it was abandoned, nor its phases and stratigraphy. Some archaeological materials collected in the area from the Visigothic period may be consistent with this chronology.

The excavation of the villa revealed 48 rooms, centralized in a small peristyle with four columns, a rather original architectural plan for this type of site [6–12]. It comprises a small, but complete, thermal complex to the west, with a second peristyle attached, several access passages with mosaics, and a large corridor leading to a monumental *natatio*, measuring 39 by 8 m [11]. Although it has never been well publicized, Pisões is a relevant villa as far as we can tell from an inscription consecrated to the goddess *Salus* by *Numerius*, slave of *Caius Atilius Cordus*, who was probably the owner of the complex by the 1st century AD. Given the importance of the villa, it is likely that there is a *pars rustica* and a *pars fructuaria* near the *pars urbana*, but it has not been detected, although three marble wine or oil presses have been documented.

The villa belongs to the University of Évora since 2017 and is part of the Experimental Farm of Almocreva. In this area, an experimental field for archeological sciences is intended

to be installed to carry out multi-disciplinary research, with applied geophysics being one of the sciences contemplated [13].

In 2017, several geophysical surveys have been carried out with different methods to prospect the subsurface and assess the state of conservation of some structures. Between 2018 and 2020, magnetic (vertical gradient mode) and ground-penetrating radar (GPR) surveys were carried out in several sectors around and inside the *pars urbana* (Figure 2) [14–16].

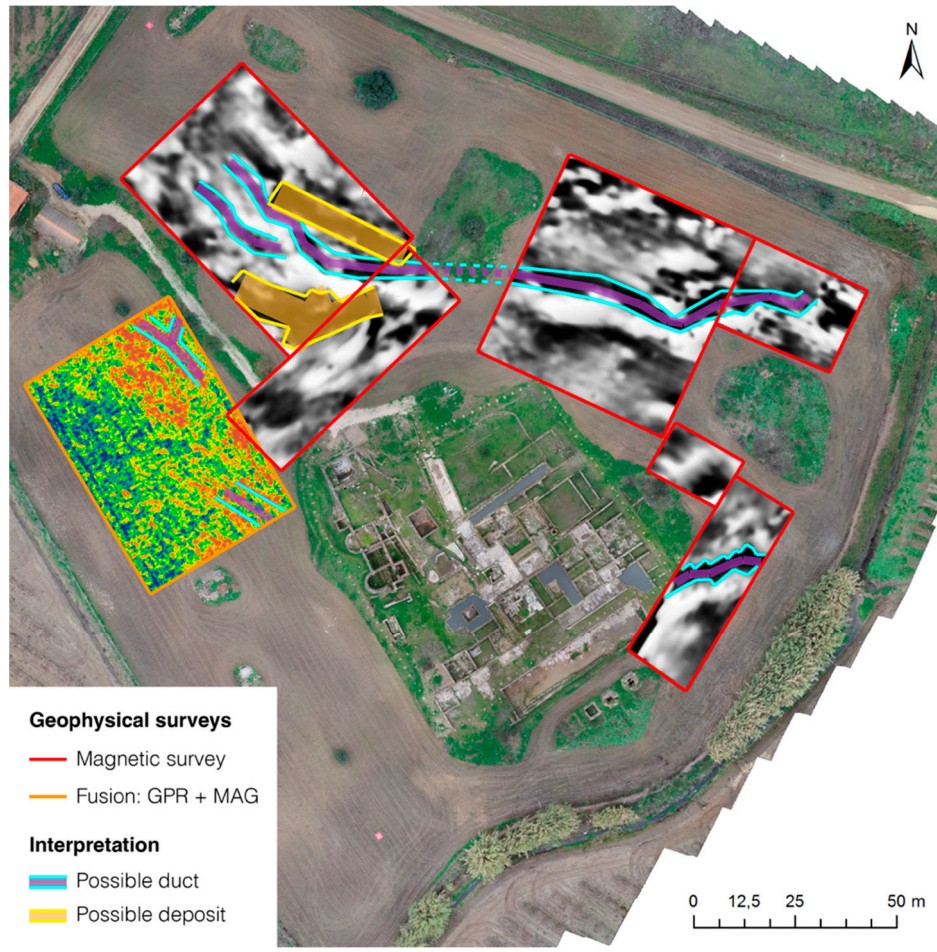

**Figure 2.** Location and results of the geophysical surveys carried out at Pisões.

In the context of this work, we would like to know how the water supply and its distribution for the hydraulic elements of the villa were articulated. To this end, we carried out an archaeological study of the emerging elements that could be analyzed and a series of ground-penetrating radar (GPR) surveys around the perimeter and inside the site. This information was georeferenced with a differential global navigation satellite system (GNSS), with centimeter precision, to determine the slope and direction of water flow. A geospatial survey with an unmanned aerial vehicle (UAV) was also carried out to obtain a photogrammetric model to study altimetric elevations and their relationship with water transport. Based on all of this, we present, in this article, the results with a proposal of water distribution, relevant information for the interpretation of GPR data, and hypotheses generated that can provide insight into the existence of other constructive phases that had not been considered.

## 2. Materials and Methods

### 2.1. Archaeological Method

The fieldwork consisted of the analysis of the emerging elements, for which we proceeded to photograph, measure, and locate with differential GNSS. All the information

was introduced in a database constructed in a GIS environment (ERSI ArcGIS) to produce all the maps located in the space, allowing the analysis and interpretation of the geophysical results related to the spatial information of the structures that would improve the understanding of the archaeological site. In the initial stage, we proceeded to register the hydraulic structures and conduits, making the first catalog proposal for their location and continuity. We know the direction of the water flow due to the accurate topography study with GNSS. The elements were classified according to their formal similarity, given that there were water conduits that had different functions, from small pipes only for draining small quantities of water to galleries and sewers into which a person can fit. The elements that were studied were photographed to determine their relationship, measured, and contrasted with the rest of the information. In this stage, a first interpretation of the site was carried out, analyzing what the distribution of water in the villa must have been like so that we could plan geophysical surveys in the form of cross-sections in the areas we estimated.

### 2.2. Geophysical Method

In this work, the GPR method was used to prospect the hydraulic pipeline system that transported water and effluents in the exposed *pars urbana*. GPR GSSI model SIR-3000 equipment was used, using two antennas with different central frequencies (400 and 1600 MHz). The acquisition parameters for each antenna are detailed in Table 1. The detection capacity for structures of a certain size is related to the central frequency of the antennas used and the depth at which the structures are located, which will influence the values of the radial and lateral resolution parameters that establish the limits from which the structures can be detected separately [17]. The values for each antenna considered are presented in Table 1. The processing applied to each B-scan using the GSSI RADAN 6.5 software is detailed in Table 2 and consisted of the following standard applications [17]: surface position correction, background noise removal, and gain rectification. The processing applied intends to highlight the hydraulic conduits, being marked in the form of hyperbolic reflectors for easier interpretation.

**Table 1.** Acquisition parameters of GPR surveys.

| Acquisition Parameters | 400 MHz Antenna | 1600 MHz Antenna |
| --- | --- | --- |
| Antenna frequency (MHz) | 400 | 1600 |
| Scans per meter | 40 | 200 |
| Samples per trace | 1024 | 1024 |
| Temporal depth (ns) | 50 | 20 |
| IIR band-pass filter (MHz) | 100–800 | 295–1930 |
| Radial resolution * (m) | 0.0438 | 0.0109 |
| Lateral resolution * (m) | 0.2092 | 0.1046 |

* Considering that the structure is located at 0.5 m depth and a velocity of 0.07 m/ns corresponding to clay soil.

**Table 2.** Processing parameters of the GPR datasets performed in the GSSI RADAN 6.5 software.

| Processing Operation | 2013 Survey | 2015 Survey |
| --- | --- | --- |
| Correction of the surface position | 4.83 ns | 2.03 ns |
| Background removal | Average all scans | |
| Gain adjustment | Exponential, 1 point | |

### 2.3. Geospatial Methods

Complementarily, different GNSS measurements were carried out to record the absolute altitude and position of the lower inner part of the ducts. EPOCH 50 (base and rover) Spectra Precision equipment was used. The processing was carried out later with correction using data from a continuous GNSS station of the National Geodesic Network of the Directorate-General for the Territory of Portugal.

A photogrammetric survey was also carried out with an UAV (DJI Phantom 4 Pro), with ground control points obtained with differential GNSS measurements.

## 3. Results

### 3.1. Detailed Analysis of Each Water Conduit

In the first stage of surface work, we were able to discover a total of 46 water conduits in the villa, within which they served different functionalities and may not have been related to each other. Figure 3 shows the visible conduits, which are described below in groups according to their function. The altitude values of the visible ducts were measured with differential GNSS (Table 3).

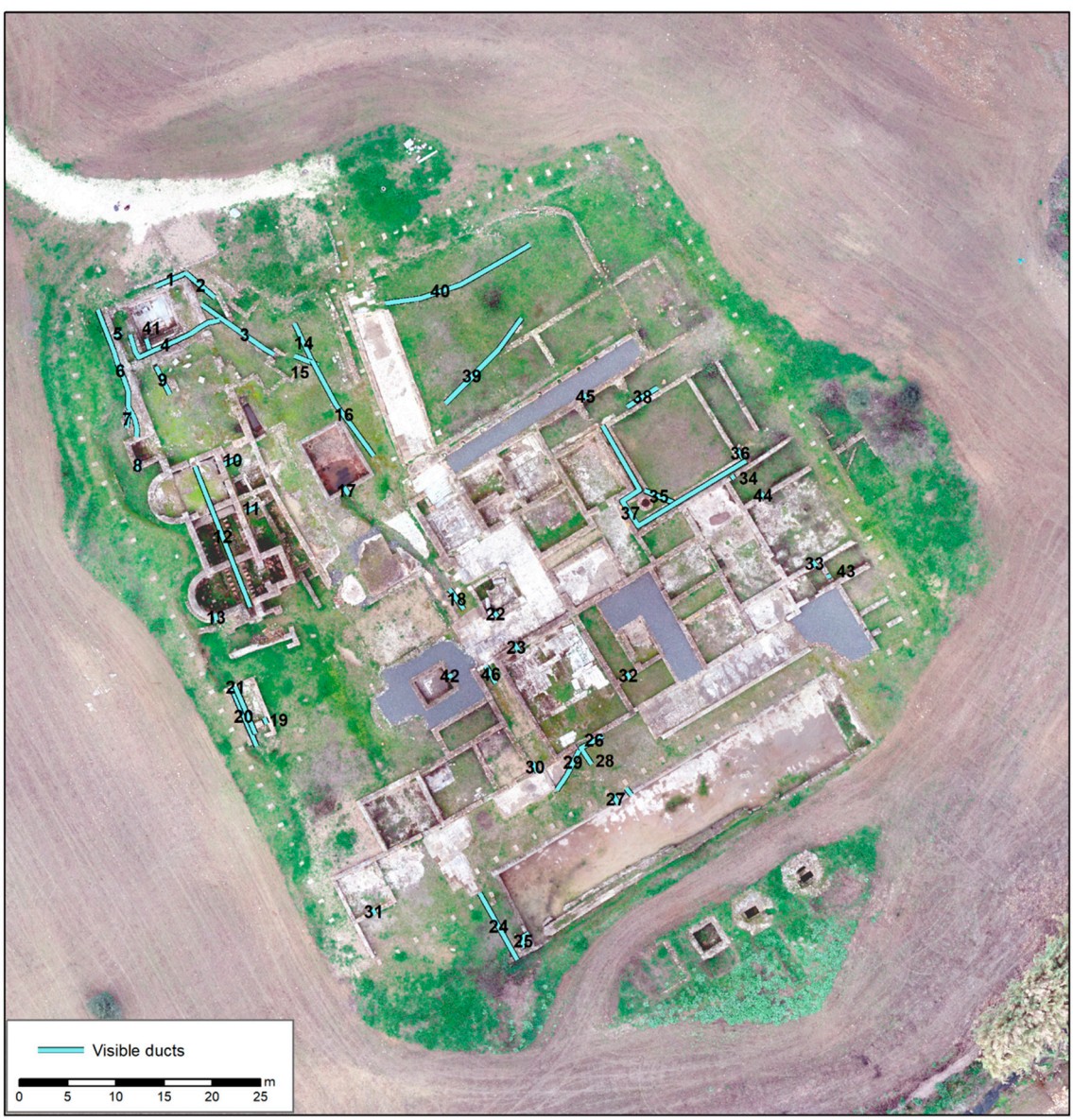

**Figure 3.** Location of visible water conduits in Pisões on the orthophotography of the site.

Ducts 1 to 9 correspond to the original supply of the thermal building, where pipe 1 is at an elevation of 181.60 m, the highest topographical point in the whole villa. This part of the complex can be understood quite well, given that water flows from pipes 1 and 2 to 6 and from there through 3, 4, and 5 to the *natatio* located there, exiting at pipe 9.

**Table 3.** Altitude values of the top of each visible water conduit.

| Conduit | Altitude Values (m) | Conduit | Altitude Values (m) |
|---|---|---|---|
| 1 | 181.60 | 24 | 176.67 |
| 2 | 181.55 | 25 | 175.90 |
| 3 | 181.20 | 26 | 177.60 |
| 4 | 180.40 | 27 | 177.20 |
| 5 | 180.42 | 28 | 177.30 |
| 6 | 180.80 | 29 | 177.30 |
| 7 | 180.50 | 30 | 177.90 |
| 8 | 179.70 | 31 | 177.40 |
| 9 | 179.48 | 32 | 177.90 |
| 10 | 179.81 | 33 | 178.10 |
| 11 | 179.40 | 34 | 178.60 |
| 12 | 178.36 | 35 | 178.90 |
| 13 | 179.50 | 36 | 179.00 |
| 14 | 180.30 | 37 | 178.90 |
| 15 | 180.12 | 38 | 179.70 |
| 16 | 180.20 | 39 | 180.14 |
| 17 | 179.38 | 40 | 180.70 |
| 18 | 178.40 | 41 | 179.40 |
| 19 | 178.40 | 42 | 178.20 |
| 20 | 178.30 | 43 | 178.10 |
| 21 | 187.90 | 44 | 178.60 |
| 22 | 178.35 | 45 | 180.30 |
| 23 | 177.95 | 46 | 180.20 |

Ducts 10 to 13 are part of the main private *therma* or *balneum*, but only 12 and 13 correspond to the water exit of the whole system, while the other two elements are isolated and at lower levels.

Ducts 14, 15, and 16 correspond to a very shallow water supply circuit that is difficult to pinpoint, as we can neither see its origin nor its end.

Ducts 17 and 18 correspond to the exit of one of the large pools and a small sewer that is visible near the center of the villa, which corresponds to pipe 46.

Elements 19, 20, and 21 are related to the latrines, a clean water inlet, and a sewage exit.

Ducts 22 and 23 are the water exit of the central *impluvium* and the water inlet of one of the rooms, which, due to its marble cladding, could be a low recreational pool.

Ducts 24 and 25 are the water exits of the monumental *natatio* and possibly another conduit that takes water from other parts of the villa to this point, which seems to be a collector.

The remaining ducts are very difficult to identify. The connection of elements 26, 27, 29, and 30 is not clear. The same is true for pipes 35, 36, 37, and 38, which seem to be related to a well located to the northeast of the villa. It is not clear whether element 40 is a wall or a conduit, but structure 39 is an aqueduct with manholes that must have supplied water to the town, given that it is located at a very high elevation. The remaining elements are isolated in the villa and, in general, do not seem to be connected to other hydraulic elements. They are mainly holes in the walls measuring 8 × 16 cm, which could well be elements for cleaning the rooms or even the restoration and consolidation of the elements themselves to prevent the accumulation of rainwater.

Figure 4 shows examples of the different elements that we detected in the villa with a water conduction function, whose numbering corresponds to the structures identified in Figure 3. Conduit 4 would be under the pavement and would serve to conduct water within the enclosure itself, while conduit 6 is on the side of the *balneum*. Duct 9 is the water exit of one of the *natatio*, and duct 12 is where the water would have gone, which can be seen on the floor of the well-preserved *hypocaustum* with its brick basins. Element 13 is the only water exit that we located on the side of the *balneum* for hot water. Structure 18 is the water conduit that would have supplied water to the center of the villa, while structure

20 is a duct coming from the latrines. Structure 23 is a strange element that seems to be to supply water to the room that could be a low *natation*. Finally, structure 25 would be the main sewage collector of the villa in a poor state of preservation.

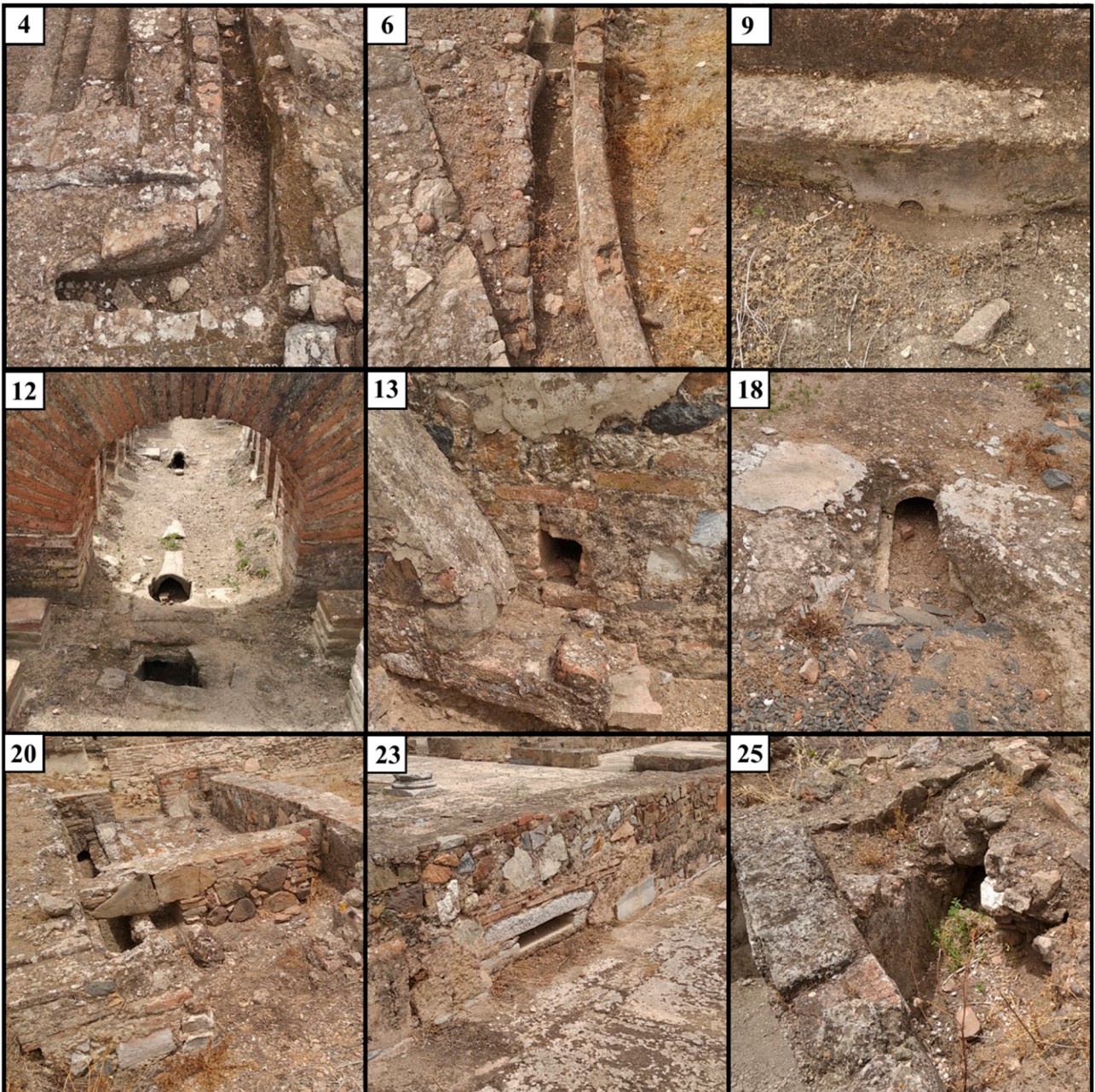

**Figure 4.** Examples of water conduits of Pisões. The numbering corresponds to the structures identified in Figure 3.

Considering all the elements analyzed, the only water supply point, i.e., where it may have come from, is conduit 39. This is a small, partially collapsed brick vault in which small access and cleaning well is preserved, as well as part of its roof being visible. This construction is very similar in shape to an aqueduct located northeast of the villa [18,19] and could be the access point for water from nearby springs. In addition, structure 39 is very similar in shape to structure 11, so all three could be part of the same aqueduct, as can be seen in Figure 5.

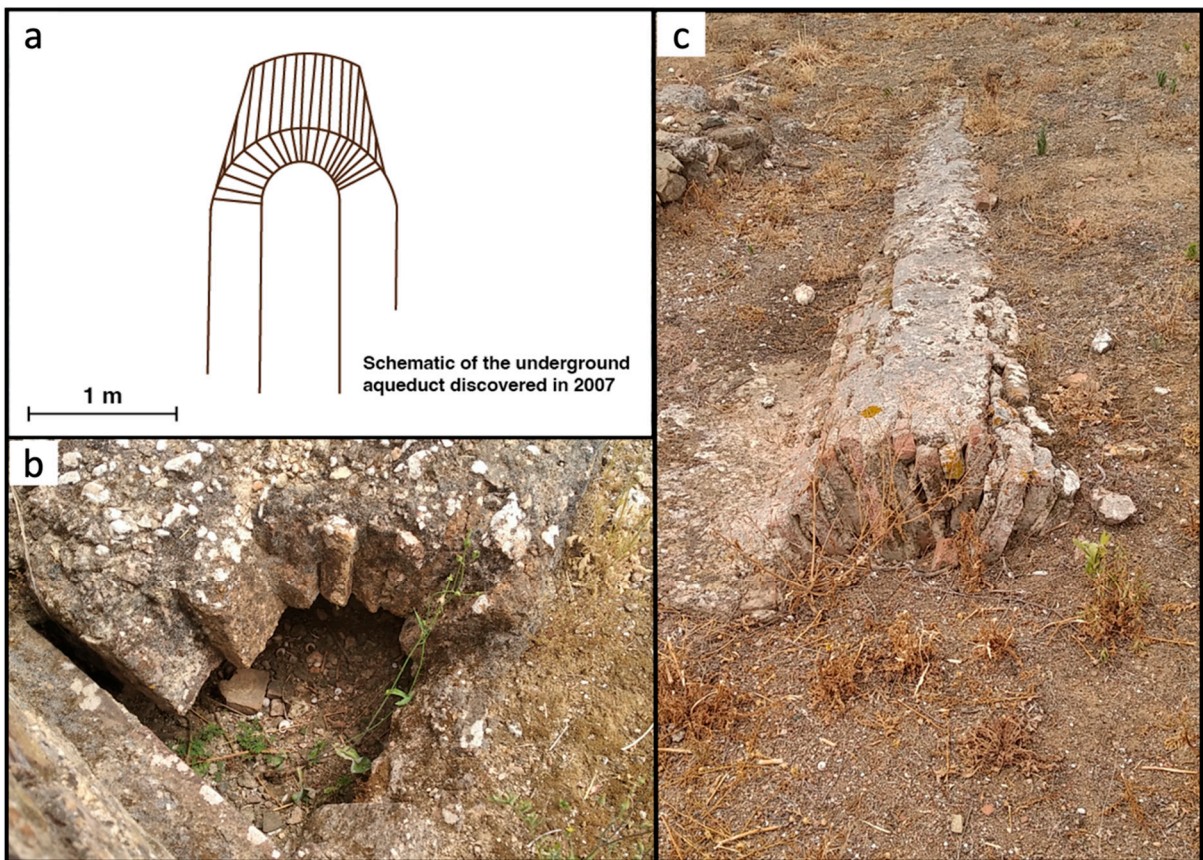

**Figure 5.** (**a**) Underground aqueduct schematics discovered in 2007 in an archaeological survey carried out on the outside of Pisões. (**b**) Upper part of duct 11. (**c**) Traces of the upper part of duct 39.

*3.2. Structure Detection Using GPR*

The results of the geophysical studies of the GPR and magnetic methods reveal that there is a great possibility for large structures to exist in all sectors that surround the excavated urban part [14,15]. For Pisões, a problem of a lack of perceptibility of buried structures was also identified, caused by the physical and chemical characteristics of the soil and the materials that made up the structures [14,15]. For the GPR method, the size of the clay particles caused scattering to add to the strong attenuation characteristics of this type of soil. The characteristic hydration of clays also caused strong attenuation. Even with these conditions described, GPR surveys were carried out using antennas of different central frequencies until it was proven again which ones allowed the best results to be obtained.

Therefore, the study carried out consisted of the acquisition of 38 B-scans around and inside the *pars urbana* using antennas with the following central frequency values: 100, 200, 400, and 1600 MHz. These frequencies are suitable to prospect buried structures in an archaeological context. However, around the excavated sector of Pisões, it was only possible to detect reflections compatible with structures with the 200 MHz antenna. Near the border of the *pars urbana*, any reflections were detected. In the inner part of the *pars urbana*, the 400 and 1600 MHz antennas allowed the detection of buried structures.

Thus, in the inner part of the *pars urbana*, 17 reflection patterns were identified in the B-scans, all compatible with water conduits, and, at expected locations, near-visible conduits or structure holes were detected. The location of these patterns is shown in Figure 6.

The results of the GPR survey can be observed in the following figures. All B-scans are overlaid into a picture of the prospected sector in Pisões, with a schematic of the profile location and direction.

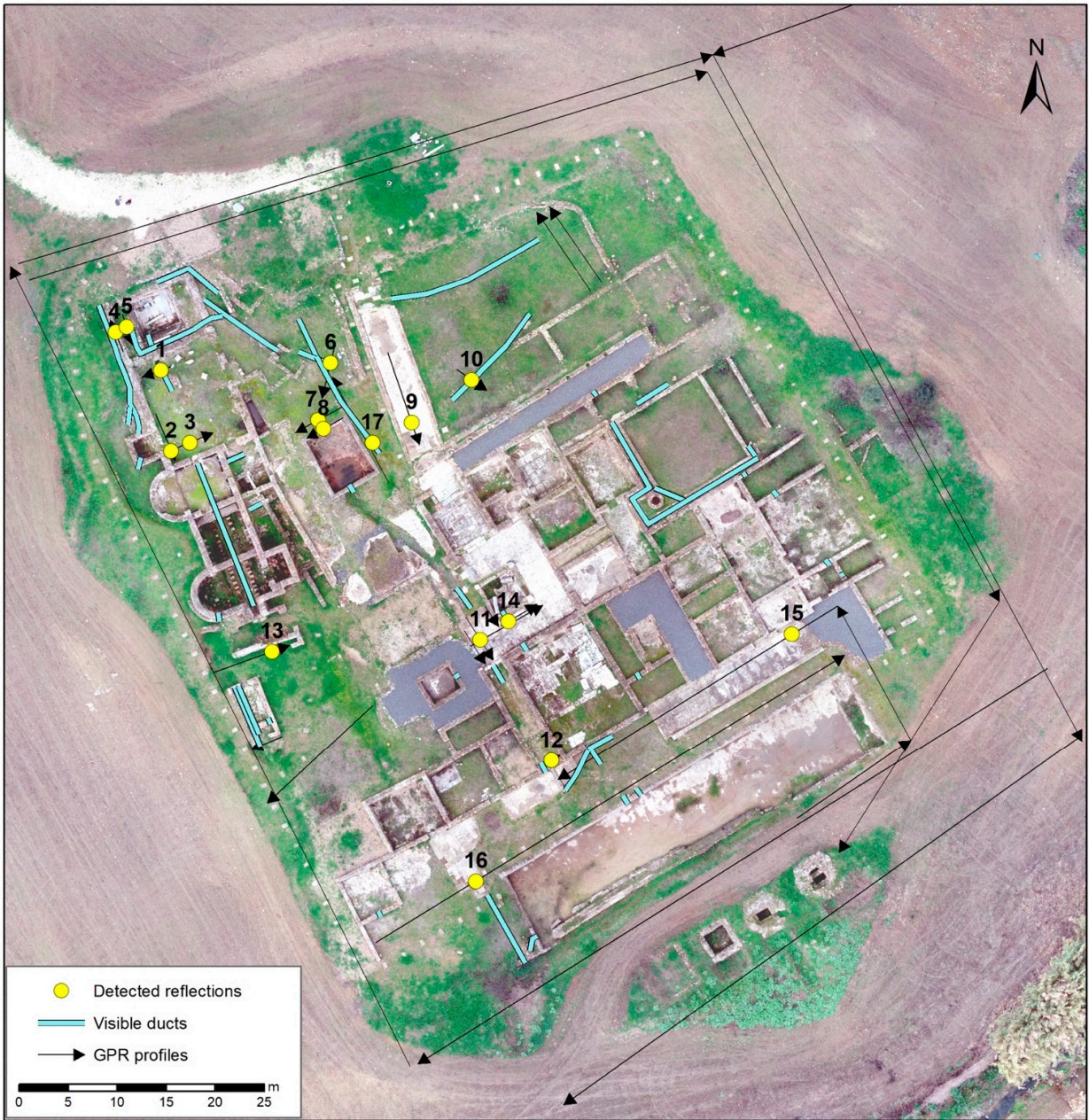

**Figure 6.** Location of the GPR reflection patterns compatible with water conduits.

In Figure 7, a B-scan acquired near the remains of two conduits is shown. However, just one can be observed and correspond to the observation in the field. As these tubes are quite degraded, it is possible that parts of them no longer exist.

In Figure 8, two B-scans carried out with 400 and 1600 MHz antennas are shown. Both profiles were acquired partially in the same location and show a reflection pattern compatible with a duct in the expected location of the *peristilum* drainpipe. For small structures, the 1600 MHz antenna allowed more details to be observed. This is related to the concept of radial and lateral resolution that establish the limits from which the structures can be detected.

Figure 9 shows the spatial distribution of an underground aqueduct. Near the P054 B-scan, we can observe the top part of the structure. The results show reflections compatible with a great conduit that passes under the mosaic floor and continues toward the small pool and the thermal baths.

Figures 10 and 11 show the B-scan reflections in the continuation of drainage conduits.

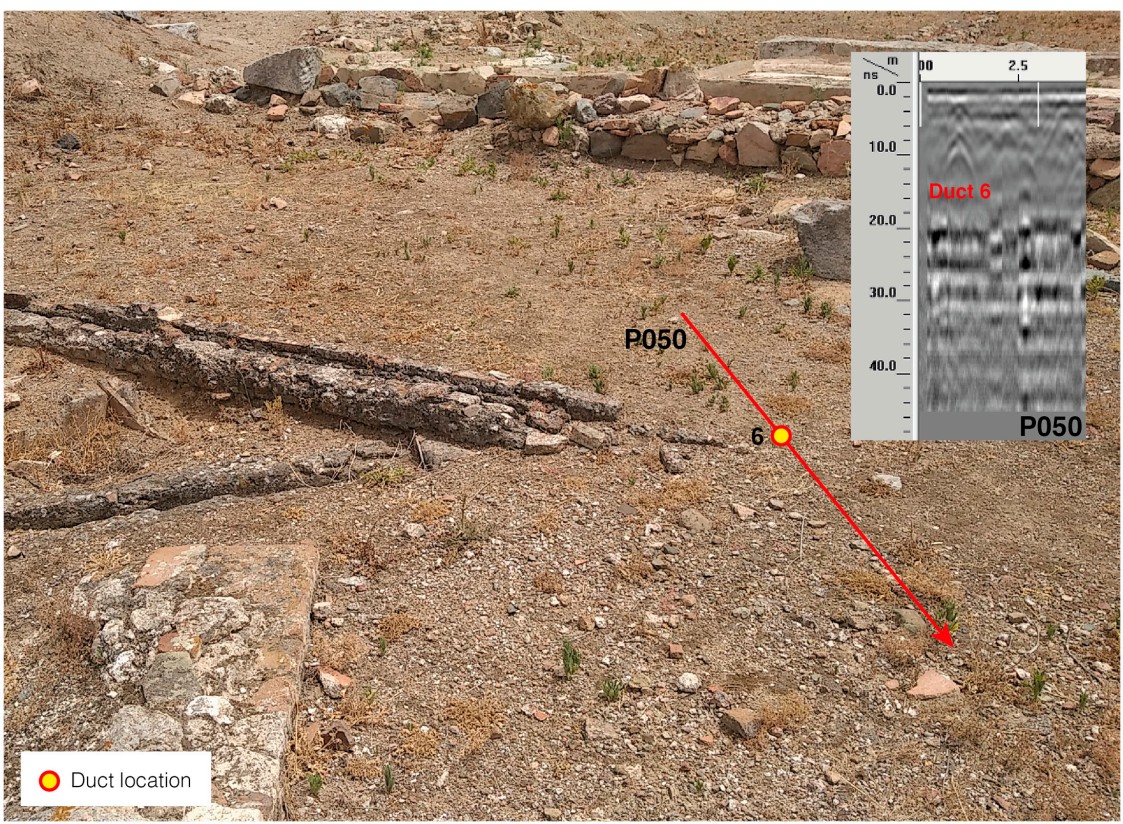

**Figure 7.** Representation of the B-scan P050 and location in the field.

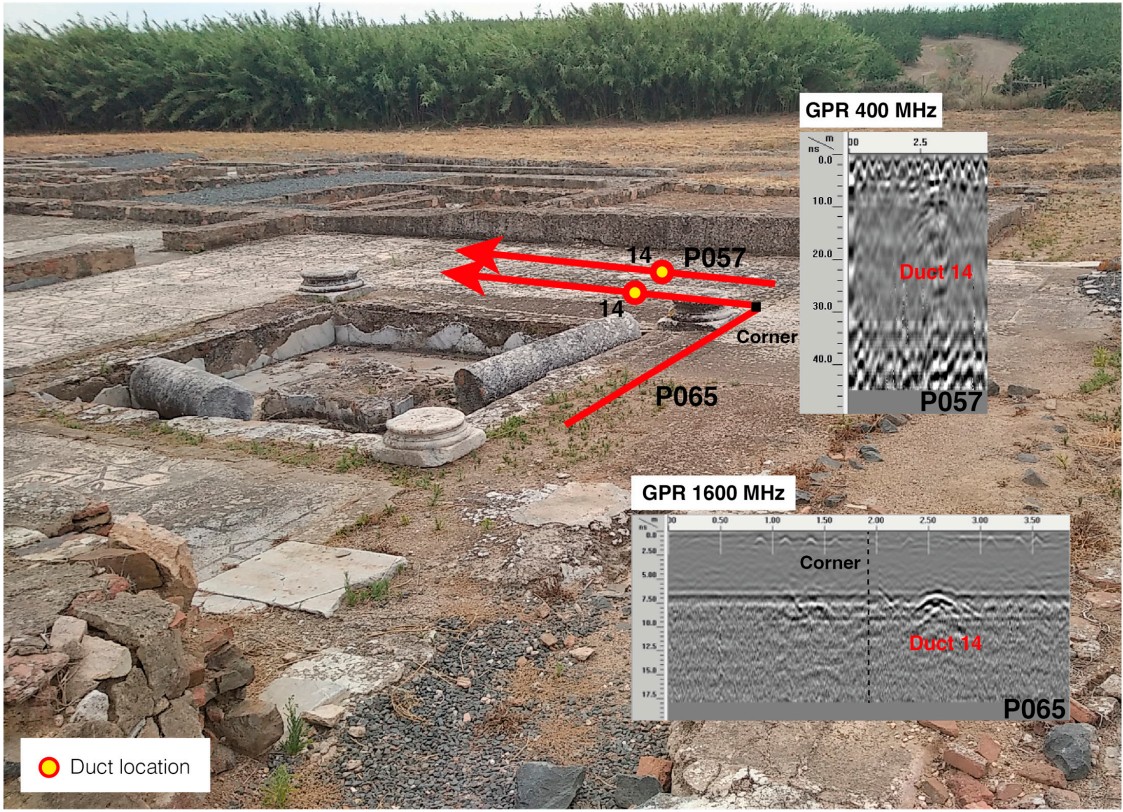

**Figure 8.** Representation of B-scans P057 and P065 and their location in the field.

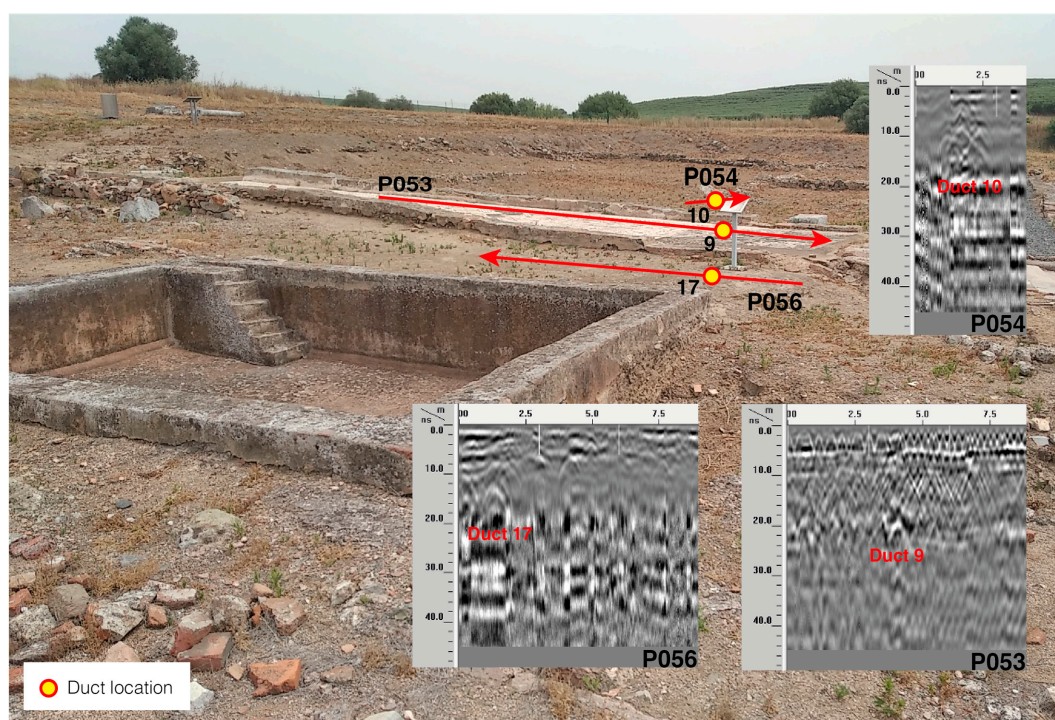

**Figure 9.** Representation of B-scans P053, P054, and P056 and their location in the field.

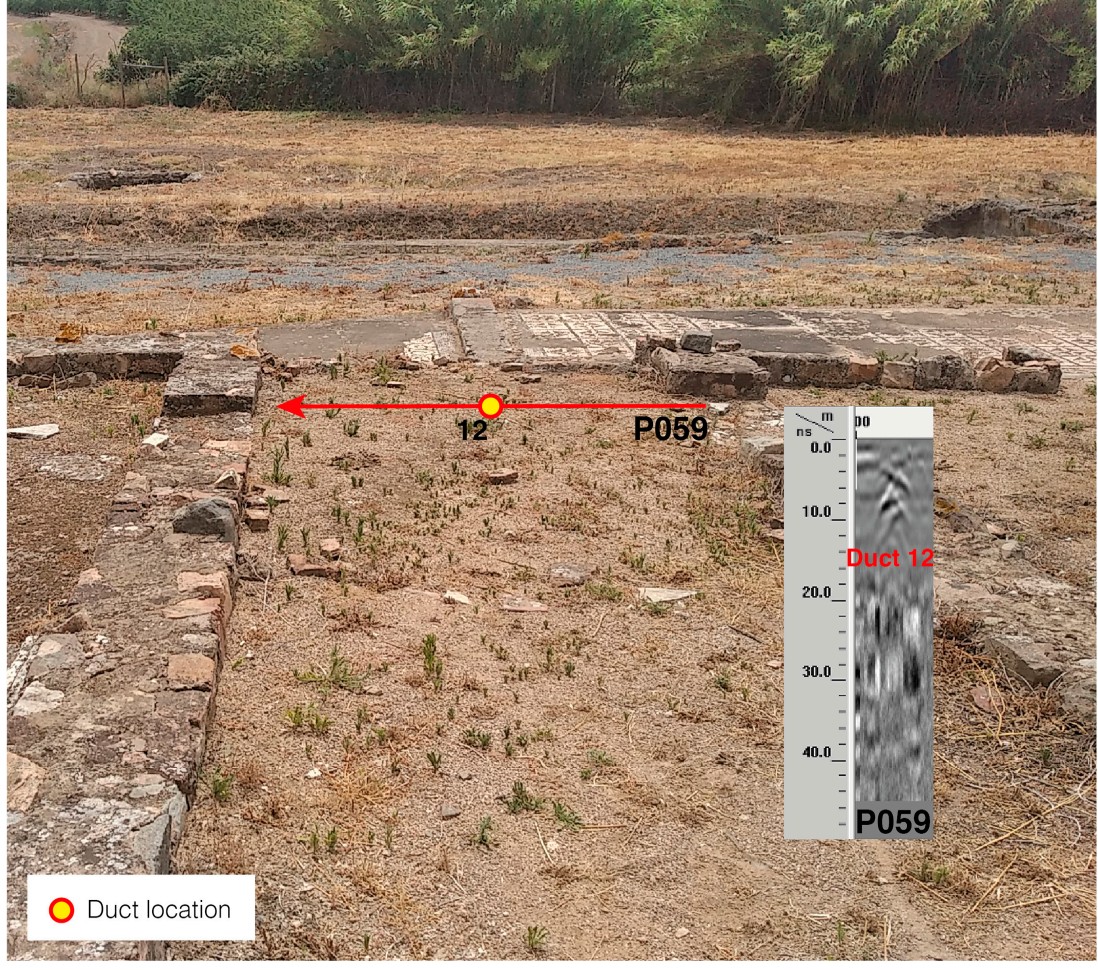

**Figure 10.** Representation of B-scan P059 and its location in the field.

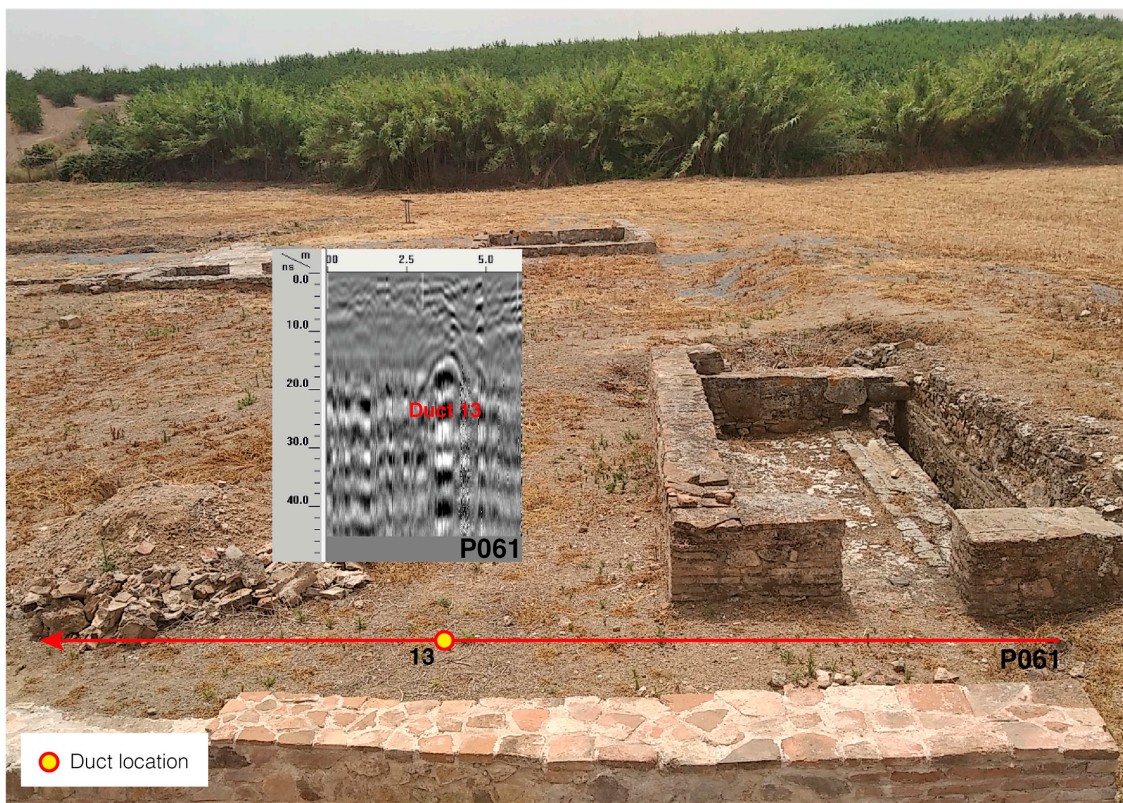

**Figure 11.** Representation of B-scan P061 and its location in the field.

## 4. Discussion

The interpretation of all information, whether archaeological, geophysical, or geospatial, allows us to make several considerations to be discussed.

The antennas used in the surveys, with central frequencies of 400 and 1600 MHz, allowed the detection of reflections corresponding to hydraulic conduit structures. Tests were carried out in places where cut ducts were observed. Differences were observed in the reflection patterns of the same structure using different antennas, caused by differences such as the investigated depth and values of radial and lateral resolution parameters, which were different for each central frequency considered and established the minimum distance between two structures from which it was possible to detect both [17].

Excluding from the initial analysis several conduits that are not related to the main entrance, such as some small ones in the rooms and others related around the well, we propose a series of conduits that should have existed, based on the combination of all the data (Figure 12). The tracing was produced by considering the connections between the visible and preserved elements that explain the water circulation of the main system. There is a high possibility that many of these features are not preserved on the surface, which may have been because of the history of the site, such as the agricultural works that destroyed structures. These problems may also have been detected in previous excavations, but the lack of technical information at that time does not allow us to know if this occurred. Therefore, we can describe the water networks that seem to have worked at the time of greater expansion of the villa when the main hydraulic elements were in operation. Added to this is a series of elements located in the northwest that have no direct connection and may date from another phase, as will be discussed later.

The detailed study between the visible water pipelines and those estimated by geophysics allowed us to reconstruct almost all the missing sections. The estimated sections are about twice the length of the parts that are currently preserved. The reconstruction of the main water connections can be observed in Figure 13.

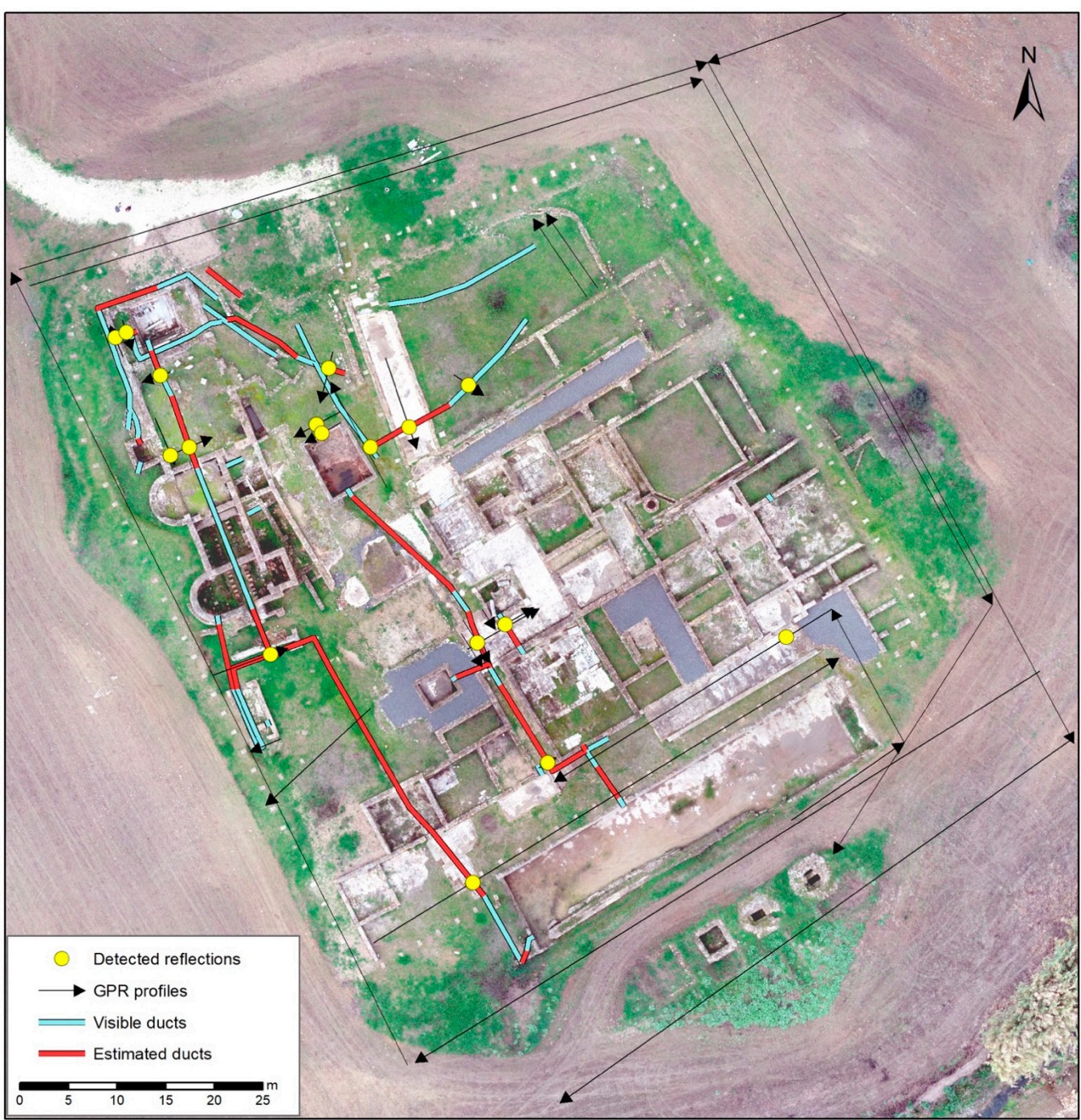

**Figure 12.** Location of estimated ducts, combining archaeological and GPR information sources.

Considering the altitude values of the bottom part of the conduits and water receptacles, we can suggest that the water access is located at the northwest corner of the *pars urbana*, with an altitude of 181.60 m, the highest value measured in Pisões. The lowest value is in the large *natatio* to the south of the villa, with an altitude value of 175.90 m. The villa is, therefore, distributed on a hill that slopes down in a south-easterly direction with a change in elevation of more than 6 m in total. The water supply must come from the northwest. This source of water to the inner part of the villa is compatible with the magnetic results shown in Figure 2 [15,16]. In this sector, anomaly alignments are visible that could correspond to conduits. Therefore, the main water access can be well reconstructed to supply the main hydraulic elements of the *balneum*, corresponding to elements 1 and 4 (Figure 13).

The first one would correspond to a *labrum* supplied from outside the building, while the second would be a cold-water *natatio*. This small pool has a hole that crosses the wall that supplies water to this structure, but there is a second conduit in continuation, confirmed by the GPR method; however, we have no further information about its distribution in space. It may be connected to *natatio* 3, which we know could have had a water conduit to supply it. This element could have been outdoors and, therefore, would not need an inlet pipe. In addition, no water passages are visible in the wall, although they may have been covered up during site restoration and consolidation.

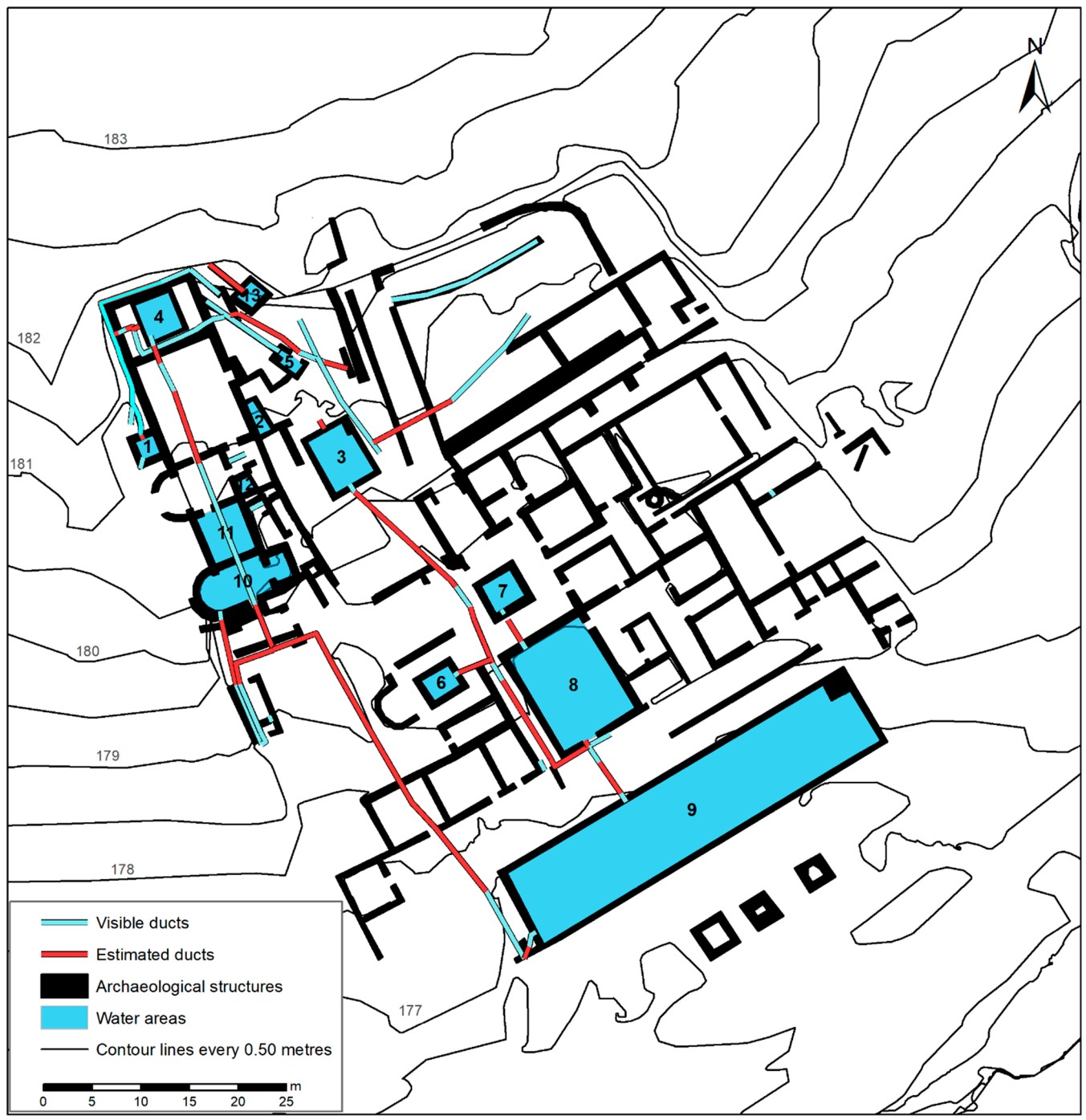

**Figure 13.** Location of the estimated water connections between the existing conduits (visible and those detected by geophysics).

Following the pipes to the *balneum*, we have no evidence of where the water enters for rooms 10 and 11, but there is some evidence that it comes from the same place as that for room 1. However, this part of the building is in a poor state of conservation. The exit from these rooms is identified in the southwest corner, after structure 10. From pool 4, passing underneath the *hypocaustum*, there is a small sewer for dirty water that connects precisely to this exit, and we have found that it also connects to the dirty water that would have come out of the latrine. We suggest that it is precisely at this point to the west that the wastewater from the entire bath complex and the latrine is concentrated. We have confirmed that the continuation of this conduit makes a bend, although the connection from there to the water collector to the south, together with the monumental *natatio*, was not detected. The geophysical data in this area do not provide clear results.

From pool 3, which has a marble sluice gate, the water flows into another conduit that crosses the main path and stores the water from *impluvium* 6 and 7. In room 7, we observed that the water does not go directly to this conduit, but to room 8, which is a low pool lined with marble. We did not detect any conduits between these two elements, but they were connected to take the water to the monumental *natatio* 9. It is possible that all these elements could be outdoors, with their distribution to the main pool and from there to the south exit collector that connects the water of these elements to the water of the *balneum*. We did not find elements implying that, in water areas 3, 6, 7, 8, and 9, there is a water supply from an aqueduct, but that it would be mainly rainwater. Even so, the entire complex could have been supplied by pool 3; therefore, this would be a water reservoir in case the impluvium ran out of water.

With all this, we have another series of isolated elements that we have not analyzed. As we have already mentioned, the well to the east has a series of conduits around it, which we believe are related to storage and cooking areas, whose water supply would be from the well itself. The same is true of the northeastern section of the villa, where there are a series of isolated conduits, such as the aqueduct mentioned above. To gain an understanding of these structures, it is necessary to locate the *pars rustica* and *pars fructuaria* of the villa, which requires the application of methodologies dedicated to this purpose [20].

Above this element, we have a series of pools that have been interpreted as part of the previous villa, specifically structures 2, 5, 12, and 13; therefore, there are two superimposed thermal complexes. This was already detected in previous studies [11] and was observed in the field and in the UAV results (Figure 14) in this study. This hypothesis includes *natatio* 3 within these elements, where there would also be a *hypocaustum* below some archaeological elements at a much lower level than the second one, something we were able to verify given that several pools were found in a profile approximately one meter below those of the second thermal complex. We also verified that the directions of the two generations of bath complex were slightly rotated from each other.

Here, we discuss the water inlet in the villa. Undoubtedly, due to the appearance of the aqueduct referred to in Figure 5, it could be the main source of water coming from nearby springs. However, the continuation of this aqueduct up to the second *balneum* must be questioned, given that the roof of the aqueduct is at an altitude of 180.10 m and 179.80 m, i.e., with a difference in height of 30 cm at the two points where we were able to document it. Given the height of this vault-shaped aqueduct, the water could have circulated at an altitude of 179.50 m, with the hydraulic elements of the *balneum* at 179.70 m at the lowest point and 180.40 m at the highest point. It is unlikely that the water was distributed from these elements, especially as the access to the water for the northernmost *natatio*, which is well preserved, is at 180.30 m. Moreover, *natatio* 3 is right in the middle of the conduit, the bottom of which is at 179.30 m, i.e., it is impossible for the aqueduct to pass under this pool. It could go around it, but we found no traces of this hydraulic element in the prospections planned precisely for this purpose.

With these considerations, we suggest that the aqueduct found at the outside part of the villa [19] corresponds to the first phase of the villa, or is at least related to the ancient *balneum*, and to *natationes* 2 and 12, which were partially divided by the second and

more recent bath complex and *hypocastum*, which has not been excavated. *Natatio* 3 must necessarily be later and possibly contemporary with the second, as it seems to cut off the access to water from the oldest *balneum*.

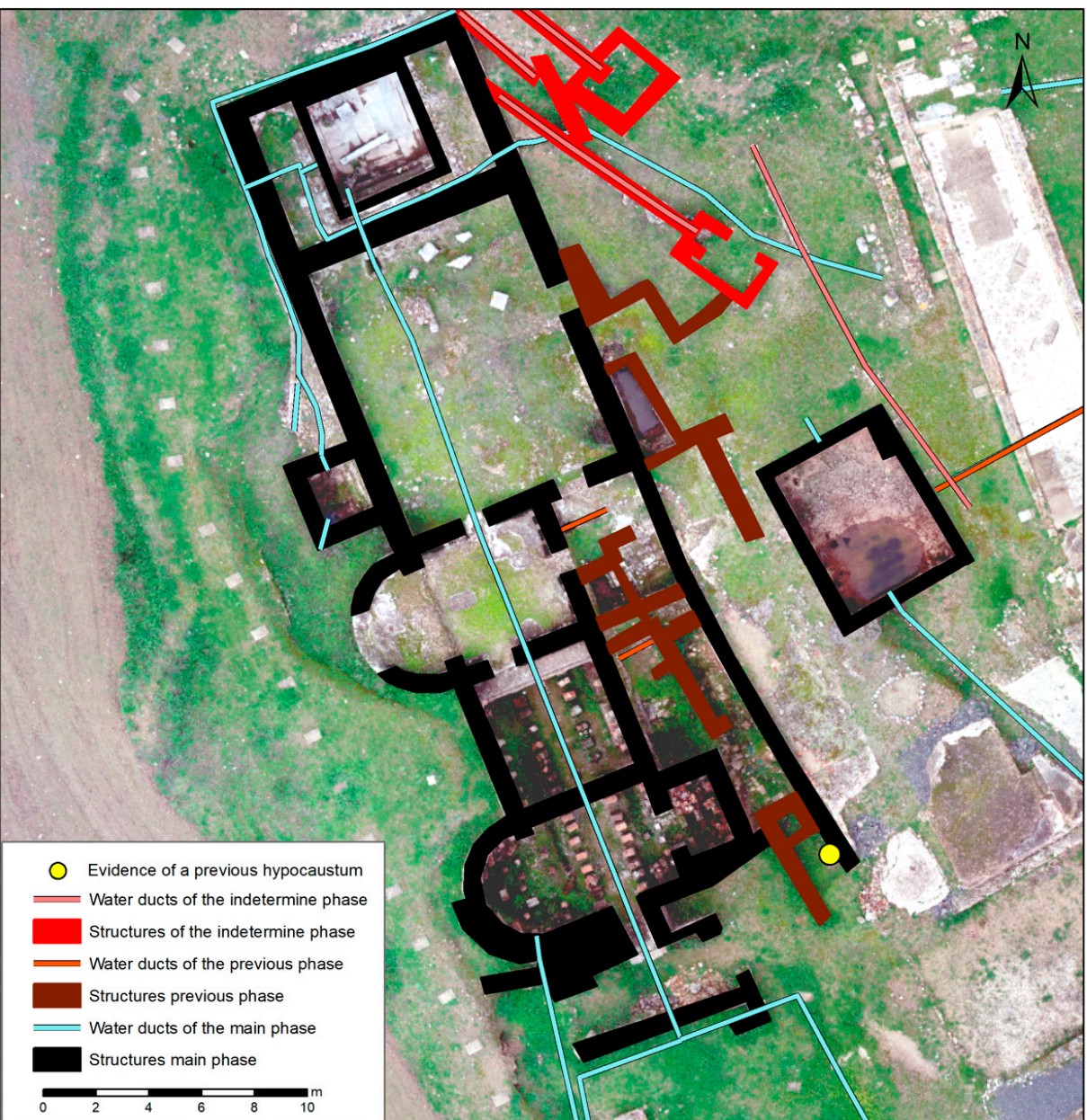

**Figure 14.** Possible different archaeological phases detected with this study.

Other elements remain, such as structures 5 and 13, which correspond to small cisterns or *natatio*, which, due to their orientation, could be related to the first phase of the villa. However, these are located at altitudes of 180.50 m and 181.60 m, making it impossible for them to have been supplied by the aqueduct. These elements have a series of conduits that have unknown directions and are not in line with the second *balneum*. Our interpretation suggests that there are three construction phases at the site. The first is the *balneum* with an aqueduct, several pools, and a partially preserved *hypocaustum*. The second is the bath complex, which is currently preserved, which was built on top of the first. Finally, in a later stage, for which we have no chronology, the water access from the *balneum* was reused to supply these other constructions, possibly for uses other than recreation. This hypothesis

is based on the impossibility of supplying water to these hydraulic elements from any of the spaces.

Regarding the origin of the water for the second phase of the villa, we consider it feasible that the main aqueduct had a branch that supplied water to the villa in the northwest sector; otherwise, there would have had to have been another element that we did not detect or was not preserved. This water deviation can be in a cistern that serves to redirect water to the second term on a higher level. In the future, this hypothesis may be confirmed by the continuation of the geophysical surveys, directed to certain areas based on the results of this work.

### 5. Conclusions

In this article, we analyzed the water distribution of an excavated villa using non-invasive technologies, such as ground-penetrating radar (GPR), complemented by datasets obtained by unnamed aerial vehicle (UAV) flights assisted by differential global navigation satellite system (GNSS) measurements. The lack of documentation in the previous excavation process makes it difficult to interpret the elements found there. Our proposal is based on an archaeological analysis of the visible elements to plan targeted geophysical surveys. The results allow us to understand a good part of the water circuit of the villa and to create criteria for determining the construction phases of the site, especially concerning the thermal complex. The process has shown the importance of the use of a wide range of frequency antennas in the GPR method. The choice of the best frequency value depends on the size and depth of the element to be detected. This approach contributes to the knowledge about the space distribution of a Roman villa in cases where documentary elements are not available, as well as provides relevant experience to understand the behavior of the GPR method in certain terrain conditions and different types of landscape.

**Author Contributions:** Conceptualization, R.J.O. and P.T.F.; methodology, R.J.O. and P.T.F.; software, R.J.O. and P.T.F.; validation, R.J.O., P.T.F., B.C., J.F.B. and A.C.; formal analysis, R.J.O. and P.T.F.; investigation, R.J.O., P.T.F., B.C., J.F.B. and A.C.; resources, R.J.O. and P.T.F.; data curation, R.J.O., P.T.F., B.C., J.F.B. and A.C.; writing—original draft preparation, R.J.O. and P.T.F.; writing—review and editing, R.J.O., P.T.F., B.C., J.F.B. and A.C.; visualization, R.J.O., P.T.F., B.C., J.F.B. and A.C.; supervision, R.J.O. and P.T.F. All authors have read and agreed to the published version of the manuscript.

**Funding:** This work has been supported by the European Union through the European Regional Development Fund included in COMPETE 2020, and through the Portuguese Foundation for Science and Technology (FCT) project UIDB/04683/2020 (Institute of Earth Sciences).

**Data Availability Statement:** Not applicable.

**Conflicts of Interest:** The authors declare no conflict of interest.

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
