# Peer review of "Studying the Water Supply System of the Roman Villa of Pisões (Beja, Portugal) Using Ground-Penetrating Radar and Geospatial Methods"

_remotesensing, doi:10.3390/rs15051447_

Round 1
Reviewer 1 Report
This work proposed a comprehensive description on the function of water elements in Roman villas using GPR and a model for the water transport inside the villa and estimate the location of the water supply. It is very interesting for me. In this paper, the author only gives the B-SCAN map of the ground penetrating radar, but the effect of the ground penetrating radar is affected by the terrain. How does the author consider this aspect. Some pictures, such as Figure 8, in the article are not very clear. It is suggested that this article be published after minor revision.
Author Response
REVISION 1
On behalf of all authors, we thank you for reading the manuscript and for the suggestions and opinions expressed in this review.
- “This work proposed a comprehensive description on the function of water elements in Roman villas using GPR and a model for the water transport inside the villa and estimate the location of the water supply. It is very interesting for me. In this paper, the author only gives the B-SCAN map of the ground penetrating radar, but the effect of the ground penetrating radar is affected by the terrain. How does the author consider this aspect. Some pictures, such as Figure 8, in the article are not very clear. It is suggested that this article be published after minor revision.”
Rui Oliveira: We are aware that the changes in the elevation of the terrain affect the comprehension of the GPR results. However, in all the B-scans were acquired in sections with conditions to apply GPR correctly, without elevation changes, plain terrain and without obstacles. Figure 8 was rectified to improve the understanding.

Reviewer 2 Report
From an archaeological and historical point of view, the article is very good and interesting, but from the point of view of the methodology and used techniques and their discussion of use and results, they seem rather weak, so this aspect should be expanded and developed more extensively. From the work of the GPR and the proposed hypothesis, the authors explain very well how the whole water ducting system worked in the villa, but due to the subject matter of this journal, the technical part of the method used should have more weight.
The following are the aspects that we consider should be taken into account in order to accept the publication of this article.
In figure 2, it would be interesting to show the magnetic survey maps without the superimposed interpretation, so that the validity and certainty of the interpretation proposed by the authors can be assessed.
In lines 100 to 112 they talk about the use of a GIS, but we wonder: where is the GIS? It would be good to know how it was developed, if that is the case. Otherwise, I do not understand what the GIS contributes to this study. In fact, it seems that the authors consider that using a GIS means that the cartographic outputs have been made in a GIS software. This is not using a GIS, nor is it justified that GIS has helped in any way in the documentation and analysis of the GPR results and GNSS measurements.
We believe that either this aspect should be explained and truly shown, or any mention of it in the article should be removed, so as not to confuse the reader who might be looking for a real use of GIS in this work.
Figure 5a should be replaced by a better quality and higher resolution figure.
In figures 6 and 12 the numbering of the profiles should be indicated to make it easier to find the location of the pipelines in figures 7 to 11.
In the text it is mentioned that there are GPR profiles that have not yielded positive results in terms of the location of ducts, and yet in figure 12 there are proposed duct layouts that should have appeared in these profiles. We consider that if these cases are not explained, we are left without knowing whether GPR fails as a suitable method for this work, or whether there are other circumstances that prevented the location and therefore the GPR technique could be considered valid.
Although in the discussion, lines 249 to 261 mention the possibility that the archaeological record has been lost due to various factors, it would be interesting to show these failed profiles, and to indicate where the conduits should have been detected. In this way the reader will have the possibility to assess the validity of the interpretation given by the authors.
If, in order to show all the profiles, the number of figures and/or pages of the article would exceed the limit of figures and/or pages, we propose that they be added to the article as supplementary documentation.
Based on the explanatory proposal set out in figures 12, 13 and 14, it is not clear how these proposals fit in with the results shown in the previous 2017 work in figure 2.
Without questioning the validity of these results, as they are not the subject of this work and therefore there are no elements available to assess them, they seem contradictory to us. The authors say that the water enters the village complex from the NW sector, where the highest altitude is documented for the ducting elements, but when it comes to relating these to the results shown in Figure 2, which do not seem to show ducts clearly leading to that point in the village. We consider that the authors should explain this aspect by trying to give coherence and to adequately relate those results in figure 2 of 2017 and those in figures 12 and 13, the subject of this article.
Both in the discussion and in the results it is mentioned that different antenna frequencies have been used to determine the most valid for the documentation and detection of the remains to be searched for. It would be useful to describe in more detail the differences between the profiles generated by each type of frequency, as well as to illustrate the reader with comparative profiles of the same area at different frequencies.
The authors conclude by saying that (lines 373 to 377):
"The choice of the best frequency value depends on the size of the element to be detected. This approach is a contribution to the knowledge about the space distribution of a Roman villa, in cases where documentary elements are not available, as well as a relevant experience to understand the behavior of GPR method in certain terrains conditions and different type of landscapes".
But the reader does not know which are the most suitable frequencies according to the type of conditions, the size of the remains, etc. We consider that this needs to be explained and developed further, to really fulfil what is stated in the conclusions.
This last aspect would be fundamental in order to provide other researchers with criteria and tools to carry out this type of analysis.
Author Response
REVISION 2
On behalf of all authors, we thank you for reading the manuscript and for the suggestions and opinions expressed in this review.
- “In figure 2, it would be interesting to show the magnetic survey maps without the superimposed interpretation, so that the validity and certainty of the interpretation proposed by the authors can be assessed.”
Rui Oliveira: Figure 2 is an interpretation of the magnetic results obtained in the Roman villa of Pisões that are published in another article. We are aware of the interesting for the reader regarding the magnetic results without interpretation, but the focus of this article in not the magnetic results. We show the interpretation of the magnetic results to show the importance of the conducts that are evidenced by these results. GPR survey were made in specific locations, inside the pars urbana, in sector where we did not perform magnetic surveys due to the impossibility to walk with the magnetometer in the middle of the excavated remains. Therefore, we keep the figure 2. In the text are included since the previous version of the manuscript the citation to the previous articles where these results are published.
- “In lines 100 to 112 they talk about the use of a GIS, but we wonder: where is the GIS? It would be good to know how it was developed, if that is the case. Otherwise, I do not understand what the GIS contributes to this study. In fact, it seems that the authors consider that using a GIS means that the cartographic outputs have been made in a GIS software. This is not using a GIS, nor is it justified that GIS has helped in any way in the documentation and analysis of the GPR results and GNSS measurements.
We believe that either this aspect should be explained and truly shown, or any mention of it in the article should be removed, so as not to confuse the reader who might be looking for a real use of GIS in this work.”
Rui Oliveira: We constructed a database in GIS environment, using the ESRI ArcGIS, to gather all the information regarding the measures of structures and the location obtained by differential GNSS. This is important to produce outputs of the results overlayed to aerial images obtained by UAV. These aspects are in tune with the definition of GIS. We do not perform spatial analysis with GIS. The text of the section 2.1 has been reformulated to clarify the use of GIS and not lead the reader to interpretations different from those intended.
- “Figure 5a should be replaced by a better quality and higher resolution figure.”
Rui Oliveira: Figure 5 was rectified. The low-quality image was replaced by a schematic of the buried aqueduct discovered in 2007.
- “In figures 6 and 12 the numbering of the profiles should be indicated to make it easier to find the location of the pipelines in figures 7 to 11.”
Rui Oliveira: We rectified the figures 7 to 11 to include the numbers of the conducts identified in the figures 6 as 12. We did not include the identification of the GPR profiles in the figures 6 and 12 due to the excess of information that this extra information cause. The focus of both images is the location of the water conducts in the space.
- “In the text it is mentioned that there are GPR profiles that have not yielded positive results in terms of the location of ducts, and yet in figure 12 there are proposed duct layouts that should have appeared in these profiles. We consider that if these cases are not explained, we are left without knowing whether GPR fails as a suitable method for this work, or whether there are other circumstances that prevented the location and therefore the GPR technique could be considered valid.
Although in the discussion, lines 249 to 261 mention the possibility that the archaeological record has been lost due to various factors, it would be interesting to show these failed profiles, and to indicate where the conduits should have been detected. In this way the reader will have the possibility to assess the validity of the interpretation given by the authors.
If, in order to show all the profiles, the number of figures and/or pages of the article would exceed the limit of figures and/or pages, we propose that they be added to the article as supplementary documentation.”
Rui Oliveira: In fact, we acquired 38 GPR profiles but in the manuscript, we only show 8. This was a choice of the authors, to show only a selection of the examples where is possible to identify the conducts. We consider that the less good examples are not crucial to understand the possible of the method to detect conducts, even if supplementary material. The method itself is not being evaluated in this manuscript and there are several articles where the thematic of the quality of the GPR results were discussed. We introduced some changes in the text to clarify the choice of the GPR profiles showed in the results. Regarding the loss of information of some ducts, in fact, the ducts were destroyed so we cannot detect it with any geophysical method. We rectified the text to clarify this aspect.
- “Based on the explanatory proposal set out in figures 12, 13 and 14, it is not clear how these proposals fit in with the results shown in the previous 2017 work in figure 2.
Without questioning the validity of these results, as they are not the subject of this work and therefore there are no elements available to assess them, they seem contradictory to us. The authors say that the water enters the village complex from the NW sector, where the highest altitude is documented for the ducting elements, but when it comes to relating these to the results shown in Figure 2, which do not seem to show ducts clearly leading to that point in the village. We consider that the authors should explain this aspect by trying to give coherence and to adequately relate those results in figure 2 of 2017 and those in figures 12 and 13, the subject of this article.”
Rui Oliveira: Our apologies but we think there was a misunderstanding caused by the lack of clear explanation on the subject in the manuscript. Between 2018 and 2020 surveys of magnetic and GPR methods were carried out. These show a series of alignments of magnetic anomalies and reflections compatible with conduit-type structures, which we associate with being for the transport of water, since they have a direction and direction cutting contour lines, which allow the conduction of water by gravity, what was used in Roman times. These results motivated the study of the Pisões hydraulic system that we present in this manuscript. Regarding the topic related to the Figure 14, during the geophysical surveys, geospatial surveys were also carried out with a UAV, which made it possible to identify that there are at least three preferred orientations of structures exposed by excavation. Several bibliographical references refer that the Romans built different types of structures with some degrees of rotation between them, so that it was clear which structures belonged to the owner of the house and which the servants used in their work. Other support buildings such as the baths, as they were not part of the main house, also rotated to some extent. Considering the observations made and reported in previous works, it was also detected that there are traces of ancient baths next to the baths currently visible, which have a slightly rotated direction. The UAV results support the observation made in the field and Figure 14 shows this evidence. The text has been reformulated so that this information is clearer for the reader.
- “Both in the discussion and in the results it is mentioned that different antenna frequencies have been used to determine the most valid for the documentation and detection of the remains to be searched for. It would be useful to describe in more detail the differences between the profiles generated by each type of frequency, as well as to illustrate the reader with comparative profiles of the same area at different frequencies.”
Rui Oliveira: We agree with your comment. Figure 8 shows a case of difference in profile results acquired at the same location with different frequency antennas. The differences are caused by characteristics of the antennas that change the vertical and horizontal resolution values. This type of difference is widely described in the literature on the method and, therefore, we have not highlighted this issue since the manuscript focuses on the results of the method as an application and does not explore the method itself. We reformulated the text to make this issue clearer for the reader, mentioning examples of detection limit values for each frequency considered.
- “The authors conclude by saying that (lines 373 to 377):
"The choice of the best frequency value depends on the size of the element to be detected. This approach is a contribution to the knowledge about the space distribution of a Roman villa, in cases where documentary elements are not available, as well as a relevant experience to understand the behavior of GPR method in certain terrains conditions and different type of landscapes".
But the reader does not know which are the most suitable frequencies according to the type of conditions, the size of the remains, etc. We consider that this needs to be explained and developed further, to really fulfil what is stated in the conclusions.
This last aspect would be fundamental in order to provide other researchers with criteria and tools to carry out this type of analysis.”
Rui Oliveira: We agree with your comment and apologize for the lack of information to the reader. Information has been added in previous sections to explain the variation of detection limits with respect to the separation distance between structures as a function of the frequency of the antennas used and the distance from the surface.

Reviewer 3 Report
The proposed study appears very interesting as it is aimed at understanding a specific archaeological aspect: the definition and functioning of the water system of an ancient Roman villa. The research is well outlined in the introductory part and presents a timely presentation of previous studies.
The combination of Geospatial positioning methodologies with GPR investigations is functional to the objective of reconstructing the water pipes as a whole, as it allows the elements in the light to be crossed with the non-visible ones identified in the subsoil. Overall, the contribution is structured in such a way as to provide an organic view of the subject matter, with figures that integrate perfectly with the text, an exposition in English that is very understandable and with punctual and pertinent references.
Publication in the current version is recommended.
Author Response
REVISION 3
On behalf of all authors, we thank you for reading the manuscript and for the opinions expressed in this review.
- “The proposed study appears very interesting as it is aimed at understanding a specific archaeological aspect: the definition and functioning of the water system of an ancient Roman villa. The research is well outlined in the introductory part and presents a timely presentation of previous studies.
The combination of Geospatial positioning methodologies with GPR investigations is functional to the objective of reconstructing the water pipes as a whole, as it allows the elements in the light to be crossed with the non-visible ones identified in the subsoil. Overall, the contribution is structured in such a way as to provide an organic view of the subject matter, with figures that integrate perfectly with the text, an exposition in English that is very understandable and with punctual and pertinent references.
Publication in the current version is recommended.”
Rui Oliveira: Thank you. We made some changes to improve the manuscript.

Round 2
Reviewer 2 Report
The corrections and explanations given by the authors satisfy my requirements and I therefore consider that the text can be published.
I would like to congratulate the authors again for this very interesting work.